# Therapeutic Application of Metal–Organic Frameworks Composed of Copper, Cobalt, and Zinc: Their Anticancer Activity and Mechanism

**DOI:** 10.3390/pharmaceutics14020378

**Published:** 2022-02-08

**Authors:** Ihn Han, Seung Ah Choi, Do Nam Lee

**Affiliations:** 1Plasma Bioscience Research Center, Applied Plasma Medicine Center, Kwangwoon University, Seoul 01897, Korea; difflvin@hotmail.com; 2Division of Pediatric Neurosurgery, Seoul National University Children’s Hospital, Seoul 03080, Korea; aiipo@snu.ac.kr; 3Biomedical Research Institute, Seoul National University, Seoul 03080, Korea; 4Ingenium College of Liberal Arts (Chemistry), Kwangwoon University, Seoul 01897, Korea

**Keywords:** metal–organic framework, degradability, anticancer, biocompatibility, apoptosis

## Abstract

Effective penetration into cells, or binding to cell membranes is an essential property of an effective nanoparticle drug delivery system (DDS). Nanoparticles are generally internalized through active transport mechanisms such as apoptosis, and cargo can be released directly into the cytoplasm. A metal–organic framework (MOF) is a network structure consisting of metal clusters connected by organic linkers with high porosity; MOFs provide a desirable combination of structural features that can be adjusted with large cargo payloads, along with Cu, Co, and Zn-MOFs, which have the chemical stability required for water-soluble use. Bioactive MOFs containing copper, cobalt, and zinc were prepared by modifying previous methods as therapeutic drugs. Their structures were characterized via PXRD, single-crystal crystallographic analysis, and FT-IR. The degradability of MOFs was measured in media such as deionized water or DPBS by PXRD, SEM, and ICP-MS. Furthermore, we investigated the anticancer activity of MOFs against the cell lines SKOV3, U87MG, and LN229, as well as their biocompatibility with normal fibroblast cells. The results show that a nanoporous 3D Cu-MOF could potentially be a promising candidate for chemoprevention and chemotherapy.

## 1. Introduction

Cancer has become a large threat to human beings in recent decades [1]. In response to evolved cancers, more efficient treatments have been developed in the areas of surgery, chemotherapy, radiotherapy, and proton-beam, molecular-targeted, and immune therapies. Among these therapies, chemotherapy is the most common treatment and predominantly utilized to treat cancer [2,3,4,5,6,7]. However, it sometimes exhibits severely undesirable side effects on normal cells, resulting in fatigue, eating disorders, vomiting, hair loss, etc. To enhance therapeutic efficiency and reduce side effects, a variety of drug vehicles have been explored to release the drug to its target within an adequate time responding to the pH, temperature, light, pressure, or magnetic field [8,9,10,11,12,13,14,15,16,17,18]. An ideal drug delivery system requires several features such as large drug loading capacity, controlled release, and high biocompatibility. Although porous inorganic oxides, organic micelles, liposomes, dendrimers, and metal nanoparticles have been introduced as new drug delivery systems, they also have some intrinsic drawbacks derived from each delivery method [19,20,21]. Many researchers have paid attention to new drug delivery systems exhibiting controlled low cytotoxicity in chemotherapy. Since 2006, metal–organic frameworks (MOFs) have attracted interest for drug delivery and theragnostic science due to their controllable pore size, easily tunable compositions, and high surface area to volume, which is advantageous for drug loading capacity and stimulus-sensitive responses [22,23,24,25,26,27]. In general, MOFs can be prepared through the self-assembly reaction of metal ions or clusters with multi-dentate organic ligands and are easily designed as ideal drug delivery candidates through interesting geometries and connectivity [28,29,30,31,32,33]. More specifically, MOFs including Cr, Fe, Zn, and Zr have been studied as drug delivery vehicles loading IBU (ibuprofen), 5-Fu (5-fluoroucil), cis-platin, caffeine, etilefrine hydrogen chloride, etc. The favorable drug loading capacities of these MOFs were accomplished through encapsulation by adsorption, or hydrogen bonding and electrostatic interaction [34,35,36,37,38,39,40,41]. For instance, a photosensitizer-loaded MOF-199, PS@MOF-199 NP, was developed for photodynamic therapy, leading to moderate photosensitization to provide an effective photodynamic therapy with enhanced therapeutic effects and minimal phototoxicity via decreased endogenous GSH levels [18]. A Cu(II)-based MOF with a 1D channel was developed for anticancer chemotherapy on OS-732 osteosarcoma cells, and this study revealed that the Cu-MOF facilitated apoptosis of the cancer cells and caused ROS accumulation on cancer cells [42]. Furthermore, MOFs can be composed of bioactive metals and ligands, which offer a therapeutic effect for various bacteria and cancers [27,28,29,30,31,32,33,34,35,36,37,38]. However, some MOFs are known to release excess metal ions which may be toxic to normal cells as well as cancer cells. Hence, there is also an interest in robust MOFs which provide intrinsic therapeutic effects without any other drug loading or decomposition releasing cytotoxic metal ions.

Herein, we present the preparation of MOFs containing glutarate (glu) and 1,2-*bis*(4-pyridyl)ethane (bpa) ligands coordinated to copper, cobalt, and zinc though modified solvothermal methods. Their stabilities were studied in physiological media, and their resulting effects on bioactivity are discussed. The anticancer properties toward various cancer cells and biocompatibility to normal cells were tested. We propose the potent possibility of therapeutic applications of robust MOFs through investigation of their cell death mechanism. In this study, the anticancer properties of the MOFs and their underlying mechanisms are reported based on in vitro experimental evidence. Taken together, this scientific evidence of MOFs with anticancer properties could facilitate future research to explore potential therapeutic targets in cancer therapy.

## 2. Experimental Methods

### 2.1. Preparation of Cu-MOF **1**, [Cu_2_(Glu)_2_(μ-bpa)]·3H_2_O

A three-dimensional nanoporous Cu-MOF **1**, formulated as [Cu_2_(Glu)_2_(bpa)]·3H_2_O, was prepared from a mixture of Cu(NO_3_)_2_·3H_2_O (0.48 g, 2.0 mmol), glutaric acid (0.26 g, 2.0 mmol), and 1,2-bis(4-pyridyl)ethane (0.18 g, 1.0 mmol) in distilled water (200 mL) containing 1.0 M NaOH (2 mL) as per previously reported literature methods [43]. In more detail, the reaction mixture was added to a high-vacuum vessel and placed in an oven at 80 °C for 48 h. After cooling to 25 °C, the obtained crystals were separated by filtration, washed with distilled water, and air dried overnight. Single-crystal crystallography, PXRD, and SEM were utilized for structural determination.

### 2.2. Preparation of Co-MOF **2**, [Co_2_(Glu)_2_(μ-bpa)_2_]·4H_2_O

Co(NO_3_)_2_·6H_2_O (0.116 g, 0.4 mmol) was mixed well with glutaric acid (0.053 g, 0.4 mmol) and 1,2-bis(4-pyridyl)ethane (0.147 g, 0.8 mmol) in 10 mL of N,N-dimethylformamide (DMF) [44]. This mixture was placed in a Teflon-lined high-pressure vessel, which was placed in an oven at 100 °C for 72 h and then cooled to 25 °C. Purple-colored crystals were collected by filtration, washed with DMF, and air dried overnight. The structure of Co-MOF **2** was determined using single-crystal crystallography, PXRD, and SEM.

### 2.3. Preparation of Zn-MOF **3**, [{Zn(H_2_O)(Glu)}_2_(μ-bpa)]

The Zn-MOF **3** was synthesized by hydrothermal reactions, using methods modified from previous reports [45]. Briefly, Zn(NO_3_)_2_· 6H_2_O (0.1 mmol), glutaric acid (0.1 mmol), and 1,2-bis(4-pyridyl)ethane (0.2 mmol) were mixed in distilled water. Subsequently, this mixture was placed in a Teflon-lined high-pressure vessel, which was placed in an oven at 80 °C for 48 h and then cooled to 25 °C. The as-prepared sample of Zn-MOF was characterized by single-crystal crystallography, PXRD, and SEM.

### 2.4. Instrumentation

The PXRD patterns of the MOFs were recorded using a Rigaku MiniFlex diffractometer (Rigaku Corp., Neu-Isenburg, Germany). FT–IR spectroscopy was performed using an INVENIO-R (Bruker, Billerica, MA, USA). The surface morphologies of the MOFs were characterized by SEM (JSM-5800F, JEOL, Tokyo, Japan). Inductively coupled plasma mass spectrometry (ICP–MS, Agilent Marker 7700, RF Generator Power 1550W, Tokyo, Japan) measurements were performed in the Seoul Center of the Korea Basic Science Institute. The fluorescence intensity was read with a microplate reader (BioTek, Winooski, VT, USA), and the stained cells were imaged by a fluorescence microscope (ECLIPSE N*i*-U, Nikon Co., Tokyo, Japan).

### 2.5. Metal Ion Release Test

To measure the metal ions released from MOFs **1–3**, 1 mL of deionized water or DPBS solution was added to 1 mg of each MOF. Four samples of each MOF were prepared and stirred for 6, 12, 24, and 48 h at 25 °C. After that time, each sample was centrifuged, and the supernatant was separated from each reaction tube. The amount of metal ions released in each sample was measured by ICP–MS. The degree of degradation was represented as the concentration of metal ions released into the medium in parts per million.

### 2.6. Metabolic Viability

The metabolic viability of the ovarian cancer SKOV3 cells, and two brain glioblastoma cell lines, U87 and LN229, was determined using alamar blue dye (DAL1025; Thermo Fisher Scientific, Waltham, MA, USA). Human dermal fibroblast (HDF) cells were used to represent normal cells in this assay. The cells were seeded in 96-well plates, keeping a cell density of 1 × 10^5^ cells in a 100 µL volume. Experiments were performed at 37 °C in triplicate or more, with a metal–organic framework (MOF) with copper (Cu-MOF), cobalt (Co-MOF), or zinc (Zn-MOF). The concentration of the MOF was selected as 1.6, 6.25, 12.5, 25, 50, and 100 µg mL^−1^. Conversion of the alamar blue dye was evaluated by determining the fluorescence emission using a BioTek plate reader, with the excitation at 540 nm and emission at 600 nm.

### 2.7. Flow Cytometric Analysis for Cell Death

To determine the levels of cell death in SKOV3, U87, and LN229 cells treated with a Cu-MOF, cells were seeded and maintained at a cell density of 2 × 10^5^ cells in each well of a 6-well plate. The cells were incubated with the Cu-MOF for 24 h, harvested, and stained with propidium iodide (PI; Sigma Aldrich, Darmstadt, Germany) and Annexin. The single-cell suspensions thus obtained were assessed and evaluated by flow cytometry. After 24 h of incubation with the Cu-MOF, cells were harvested, washed with cold PBS, and then centrifuged to form a pellet. The pellet was resuspended in PI and Annexin staining solution (5 µg mL^−1^ PI) and incubated for 15 min on ice. Immunogenic cell death was analyzed by a flow cytometer using the abovementioned procedure.

### 2.8. Statistical Analysis

All quantitative data (*n* = 4) are expressed as the mean ± standard deviation (S.D.), and significant mean differences were estimated via Student’s *t*-test. Statistical significance was set at *p* < 0.05, ** *p* < 0.01, and *** *p* < 0.001.

## 3. Results and Discussion

### 3.1. Physical Characterizations of MOFs, 1–3

Three MOFs, **1**−**3**, containing glu and bpa ligands coordinated to copper, cobalt, and zinc were synthesized by various modified methods shown in Figure 1 and Table 1 [43,44,45].

The nanoporous 3D Cu-MOF **1** was synthesized by a hydrothermal reaction [43] and formulated as [Cu_2_(glu)_2_(μ-bpa)]·3(H_2_O), which contains paddlewheel Cu_2_ dinuclear units bridged by glu ligands to form 2D sheets. These sheets were linked by bpa ligands to build a 3D framework composed of well-defined 1D channels with void volumes. The geometry of Cu (II) ions in the Cu-MOF is a square pyramid coordinated with four equatorial O atoms of carboxylate and an axial N atom of bpa.

Co-MOF **2** was synthesized by a previously reported solvothermal reaction in DMF and water [44]. Glu ligands bridged Co^II^ ions to form 2D sheets, and the sheets were bridged by bpa ligands to form an infinite 3D framework. This 3D framework was two-fold interpenetrated, there was no significant void volume in the Co-MOF, and it was formulated as [Co_2_(Glu)_2_(bpa)]·4H_2_O. The coordination geometry of the Co (II) ion is a distorted octahedral arrangement, constructed by one bridging carboxylate O atom, two chelating O atoms of carboxylate, one mono-dentate O atom of glu, and two N atoms of bpa.

Zn-MOF **3** was previously prepared by the layer diffusion method [45], but it was more efficiently synthesized by hydrothermal reactions in this instance. Zn-MOF **3** was formulated as [{Zn(H_2_O)(Glu)}_2_(μ-bpa)]. Glu ligands bridge Zn(II) ions in asymmetric chelating and mono-dentate coordination modes, to form 1D chains that are further connected by bpa ligands to form a 2D sheet. In detail, water solvent molecules between sheets connect those undulated 2D sheets through hydrogen bonds to form a 3D-like framework. That is, the Zn (II) ion in the Zn-MOF is penta-coordinated, bound by two oxygen atoms from a chelating carboxylate, one oxygen atom from a mono-dentate carboxylate, one nitrogen atom from bpa, and one oxygen atom from water (Figure 1).

The selective bond lengths for the three MOFs are summarized in Table 2. In detail, the metal–oxygen bond (M–O) increased in the order of Zn–O (1.957(4) Å) < Cu–O (1.969(3)–1.993(3) Å) < Co–O (1.994(4)–2.286(4) Å). Metal–nitrogen (M–N), C–O, and N–C bonds also showed the same tendency as the M–O bond. That is, the Zn-MOF was composed of frameworks with stable bonds.

As shown in Figure 2, the purities of the as-prepared MOFs **1−3** were confirmed by PXRD patterns. These patterns were in good agreement with the simulated patterns based on the X-ray crystallographic data.

Furthermore, the FT–IR spectra of the three MOFs showed the characteristic C–O symmetric and asymmetric stretching bands at 1450–1600 cm^−1^ representing the carbonyl groups of glutarate ligands coordinated to the MOFs (Figure 3).

### 3.2. The Degradability of MOFs in Physiological Media

To investigate the participation of metal ions released from the three MOFs in the anticancer mechanism in physiological media, the MOFs were immersed in deionized water or Dulbeccos’s PBS solution (DPBS) for one day, and then their stabilities were measured by PXRD, SEM, and ICP–MS.

At first, the crystalline natures of the as-prepared MOFs were compared with those of the samples immersed in deionized water or Dulbeccos’s PBS (DPBS) using PXRD. The PXRD patterns of the Cu-MOF immersed in deionized water or DPBS solution did not present any distinguished difference from the as-prepared Cu-MOF, as shown in Figure 4. The PXRD patterns of two samples were in line with the as-prepared pattern, which indicated that the initial crystal structure was maintained after immersion in deionized water or DPBS. However, a new peak at 22.5° was observed after immersion in DPBS, which may be derived from the interaction of Cu (II) ions on the surface of the MOF with phosphate ions within the DPBS, while the initial crystalline structure was still maintained [46,47]. As a result, the Cu-MOF was expected to be considerably stable in an aqueous solution for one day.

In the case of the Co-MOF immersed in deionized water or DPBS solution (DPBS) for one day, the initial PXRD patterns of the Co-MOF were maintained without newly appearing peaks.

However, a sharp peak appeared at 41° on the Zn-MOF immersed in deionized water for one day, while the Zn-MOF was stable in DPBS solution (DPBS).

The morphological changes of the MOFs before and after immersion in deionized water or DPBS for one day were observed using SEM. As shown in Figure 5a, the as-prepared Cu-MOF was obtained as a large crystal. The Cu-MOF was broken into small pieces after immersion in deionized water for one day, but no changes on the surface were observed (Figure 5b). In the case of DPBS, the size of the Cu-MOF was maintained, but flower-like structures were newly formed on the surface (Figure 5c). This interesting phenomenon may be attributed to a reaction between Cu(II) ions on the surface and the phosphate (PO_4_^3−^), hydrophosphate (HPO_4_^2−^), or dihydrophosphate anions (H_2_PO_4_^−^) within PBS, which did not affect the crystal framework [48].

The crystal morphology of the Co-MOF immersed in deionized H_2_O showed a remarkable difference, that is, a partially dissolved shape compared to the as-prepared Co-MOF. In contrast, when it was immersed in DPBS solution for one day, its surface transformed to small flower-type shapes with various sizes. This result also suggests that phosphate anions partially coated the surface of the Co-MOF.

The Zn-MOF showed the same tendency for the size effect as the Cu-MOF, but flowers composed of long petals coated its surface after immersion in DPBS (Figure 5c).

To study the degree of degradability, the amount of Cu (II) ions released from the Cu-MOF was quantitatively measured using ICP–MS (Figure 6). After immersion of 1 mg of sample in 1 mL of deionized water for 24 h and 48 h, the amount of released Cu(II) ions was measured. Cu(II) ions were released at concentrations of 5.0 µg mL^−1^ and 3.6 µg mL^−1^ after 24 h and 48 h, respectively. On the other hand, the concentrations of Cu(II) ions released in DPBS for the same duration were only 0.1 and 0.7 µg mL^−1^, which were negligible compared to those in deionized water. Thus, due to the flower-like coating on the surface, the Cu-MOF continued to stay stable for 48 h in the physiological medium. These results agree well with the SEM images.

The Co-MOF released Co (II) ions at a concentration of 174.8 µg mL^−1^ for 48 h in deionized water. However, the amount of Co (II) ions significantly reduced by 48 h to 8.1 µg mL^−1^ in DPBS.

In the case of the Zn-MOF, it released metal ions at 89.7 µg mL^−1^ for 24 h and then 70.3 µg mL^−1^ after 48 h in deionized water. The Zn-MOF was stable, comparable to the Cu-MOF in DPBS.

These results show that the Cu-MOF and the Zn-MOF were considerably stable in physiological media, more so than the Co-MOF, and suggest the possibility of a Cu-MOF directly contacting bacteria or cancer cells, resulting in cell death. Further, the high stability of all MOFs is expected to have a positive effect on biocompatibility due to the low release of metal ions.

### 3.3. Effect of MOF on Cell Viability and Cytotoxicity

In this work, the anticancer activity of MOFs in ovarian cancer SKOV3 cells and dermal HDF cells was detected using alamar blue dye following incubation with a MOF for 24 h. The cell viability was attained using alamar blue dye. The concentration-dependent behaviors of SKOV3 were evaluated after 24 h incubation with the Co-MOF, Zn-MOF, and Cu-MOF, as shown in Figure 7. Based on the obtained results, the Cu-MOF showed a higher anticancer activity in SKOV3 cells compared to the Zn-MOF and Co-MOF. The cell viability continued to decrease as the concentration of the Cu-MOF increased. A high concentration of the Cu-MOF led to a significantly lower viability than the control. Notably, no significant changes were observed in the Co-MOF and Zn-MOF, as shown in Figure 7. For the safety assessment, we performed the cytotoxic test on HDF cells. In these results, no significant differences were observed between the three MOFs following exposure in 1 mg mL^−1^, representing a relatively high concentration (Figure 7). This could guarantee the safe treatment of cancer patients in the future. The Cu-MOF was selected for further investigations on U87 and LN229 cells with varying concentrations (1, 10, 100 µg mL^−1^) of the Cu-MOF. The U87 and LN229 cells showed a significant decrease at 100 µg mL^−1^ of the Cu-MOF after 24 h incubation, as shown in Figure 8.

We have previously investigated the occurrence of ROS due to metal ions and the oxidative reaction with the cell membrane of the bacteria [43]. In the previous study, we proposed the main cause of cell membrane collapse might be derived from the combination of radicals and hydrogen ions in the cell membrane to eliminate hydrogen ions from the cell membrane. Combined with cytotoxicity and apoptosis assays, where the MOFs were treated and reacted under physiological conditions at 37 °C in a CO_2_ incubator, the reduction cycle between Cu^2+^ and Cu^+^ on the surface of the Cu-MOF attached to cell membranes was accelerated, and the pathways that produce reactive oxygen species (ROS) changed, increasing ROS to a level that causes cytotoxicity to cancer cells and improvement of the IC_50_ [42,49]. Taken together, these results demonstrate the well-established phenomenon of Cu-MOF-activated ROS generation via chemical reaction in an aqueous solution, inactivating cancer cells, such as U87MG, LN229, and SKOV3, as well as *E. coli*, *S. aureus*, *P. pneumoniae*, and *P. aeruginosa* and MRSA. Therefore, the Cu-MOF has a good stability and reusability suitable for biomedical applications.

### 3.4. Cell Death Analysis

The effect of the Cu-MOF on cell death in SKOV3, U87, and LN229 cells was observed. To investigate whether the Cu-MOF led to the death of SKOV3, U87, and LN229 cells, the cells were seeded in a 6-well plate and then incubated with the Cu-MOF for 24 h. The cell death induced by the Cu-MOF within 24 h of incubation was determined by the uptake of the PI solution, which is indicative of a dysfunctional plasma membrane of cells. As shown in Figure 9, the scatter plot indicated no changes in the control (without a Cu-MOF) in SKOV3, U87, and LN229 cells. On the other hand, significant apoptotic cell death was observed in cancer cells with a MOF at the concentration of 12.5 µg mL^−1^, as shown in Figure 9.

In conclusion, the three MOFs had different structures in terms of bond strength, porosity, and dimension. Their bioactivity increased comparatively with their stability; meanwhile, cytotoxicity decreased as the released metal ion amount increased. Interestingly, the anticancer property was the best on the robust Cu-MOF. The Cu metal ions on the surface of the MOF are supposed to have a greater impact on cell membranes compared to Co (II) or Zn (II) ions. This result shows that the unique natures of metal ions bonded onto the surface of a framework are very important for the viability of cancer cells.

### 3.5. Summary

Three MOFs composed of Cu, Co, and Zn were prepared by a modified solvothermal reaction under high pressure. Their structures were determined by single-crystal crystallography. Their stabilities in physiological media were investigated by PXRD, SEM, and ICP-MS. Although the Cu-MOF was very stable in physiological media, it showed a high anticancer activity toward ovarian cancer cells due to the unique nature of Cu (II) ions on the surface of the MOF, such as the occurrence of ROS due to metal ions resulting from the redox reaction with the cell membranes of cancer cells; where SKOV3 cells were treated with a low concentration of the Cu-MOF, the IC_50_ value was measured by 12.5 µmol·mL^−1^ of the Cu-MOF. Further, this study shows that the cancer cell death mechanisms induced by the Cu-MOF were mediated by apoptosis. We expect this research to be valuable in developing a MOF which satisfies both biocompatibility and a therapeutic effect toward cancer, inspiring the design of various functional MOFs suitable for cancer therapy.

## Data Availability

Not applicable.

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
