# Peer review of "Therapeutic Application of Metal–Organic Frameworks Composed of Copper, Cobalt, and Zinc: Their Anticancer Activity and Mechanism"

_pharmaceutics, 2022, doi:10.3390/pharmaceutics14020378_

Round 1

Reviewer 1 Report

Han et al. have prepared three transition metal-based MOFs and their anticancer activity and mechanism. The experimental is carefully conducted, and the results have been presented correctly, and the contents fall well into the scope of the journal. However, some points need to be addressed/answered before further considered:

  1. Why the figures of the paper seem seem to fuzzy?
  2. Figure 1 - the metal ions and coordinated atoms should be labeled and their symmetry codes should be added.
  3. The characterizations of IR and TGA of three compounds should be added in paper.
  4. The formulated of three MOFs should be consistent with TGA experimental result.
  5. The reference can be to consult involving the above questions: J. Mater. Chem. A, 2020, 8, 11933. And please cited this reference.

Author Response

Review 1.

Comments and suggestions for authors

Han et al have prepared three transition metal-based MOFs and their anticancer activity and mechanism.the experimental is carefully conducted , and the results have been presented correctly, and the contents fall well into the scope of the journal. However, some points need to be addressed/answered before further considered:

  1. Why the figures of the paper seem to fuzzy?

Response: We would like to thank the reviewer for this valuable comment. We increased the resolution of all figures as you pointed out. 

  1. Figure 1 –the metal ions and coordinated atoms should be leballed and their symmetry code should be added.

Response: We would like to thank the reviewer for comment. We already labelled the metal ions and coordinated atoms at description of Figure1 like this “The color codes: green, metal (Cu, Co, and Zn); red, oxygen; blue, nitrogen; and grey, carbon” and symmetry code was described as C 2/c, P 21/n and P 21/c, respectively at space group row of Table 1. We added the comments on their symmetry as following as

“Three MOFs 1−3 containing glu and bpa ligands coordinated to copper, cobalt, and zinc were synthesized by various modified methods shown in Scheme 1, respectively [43–45].

Nanoporous 3D Cu-MOF 1 was synthesized by hydrothermal reaction [43] and formulated as [Cu2(glu)2(μ-bpa)]·3(H2O), which contains paddlewheel Cu2 dinuclear units bridged by glu ligands to form 2D sheets. These sheets were linked by bpa ligands to build a 3D framework composed of well-defined 1D channels with void volumes. The geometry of Cu (II) ion in Cu-MOF is a square pyramid coordinated with four equatorial O atoms of carboxylate and an axial N atom of bpa.

Co-MOF 2 was synthesized by a previously reported solvothermal reaction in DMF and water [44]. Glu ligands bridged CoII ions to form 2D sheets, and the sheets were bridged by bpa ligands to form an infinite 3D framework. This 3D framework was two-fold interpenetrated, there was no significant void volume in Co-MOF, and it was formulated as [Co2(Glu)2(bpa)]·4H2O. The coordination geometry of Co (II) ion is distorted octahedral constructed by one bridging carboxylate O atom, two chelating O atoms of carboxylate, one mono-dentate O atom of glu and two N atoms of bpa.

Zn-MOF 3 was previously prepared by the layer diffusion method [45], but it was more efficiently synthesized by hydrothermal reactions this time. Zn-MOF 3 was formulated as [{Zn(H2O)(Glu)}2(μ-bpa)]. Glu ligands bridge Zn(II) ions in asymmetric chelating and mono-dentate coordination modes to form 1D chains that were further connected by bpa ligand to form a 2D sheet. In detail, water solvent molecules between sheets connect those undulated 2D sheets through hydrogen bonds to form a 3D-like framework. That is, Zn (II) ion in Zn-MOF is penta-coordinated, bound by two oxygen atoms from a chelating carboxylate, one oxygen atom from a mono-dentate carboxylate, one nitrogen atom from bpa, and one oxygen atom from water (Figure 1). Further, crystallographic data showed 3 D Cu-MOF 1, Co-MOF 2, and Zn-MOF 3 were crystallized in the monoclinic space group (C2/c) (P21/n), and (P21/c), respectively (Table 1)”

  1. The chacracterization of FTIR and TGA of three compounds should be added in paper.

Response: We thank the reviewer for valuable comment and added new FTIR data in Figure 3 and TGA data were reported on previous reports as following as

-FT-IR spectrum

Figure 3. FT-IR spectra of Cu-MOF (black), Co-MOF (blue), and Zn-MOF (red)

“Furthermore, the FTIR spectra of three MOFs showed the characteristic C–O symmetric and asymmetric stretching bands at 1450–1600 cm−1 representing the carbonyl groups of glutarate ligand coordinated to MOFs (Figure 3).”

  • TGA of Cu-MOF was already reported at supporting information of ref 35, Dalton Trans. 2019, 48, 8084–8093.
  •  

  • TGA of Co-MOF was already reported at supporting information of ref 43, Sci. Rep.    2019, 9, 14983.

  • TGA of Zn-MOF was already reported at supporting information of ref 44, Cryst. Growth Des. 2013, 13, 4815–4823.

The thermal stability of these MOF does not show much difference except the amount of water staying in framework. Therefore, we considered it is not necessary to discuss their stability at high temperature in here because anticancer reaction proceeds in aqueous environment at low temperature, 37°C.

  • However, I referred J. Mater. Chem. A. 2020,8, 11933 as ref 41 for application of MOF.

Reviewer 2 Report

The article “Therapeutic Application of Metal–Organic Frameworks Composed of Copper, Cobalt, and Zinc: Their Anticancer Activity and Mechanism” by I. Han, . Ah Choi and D. N. Lee deals with the treatment of tumor cells using MOF and is based on well-known state-of-the-art.

The idea of MOF in drug delivery is very compelling, but many aspects still need to be improved before it can really be applied. In particular the work describes three MOFs based on copper, cobalt and zinc. Each metal is linked by glutamate forming a 2D sheet structure and then these sheets formed a bridge with the bpa, resulting in the 3D structure.

From the values obtained on the stability studies, Cu-MOF was stable for 24h mainly in deionized water, Co-MOF both in water and in DPBS and Zn-MOF mainly in DPBS. Evaluating in particular the results obtained on the release of the metal in an aqueous environment and in DPBS, a very low quantity of metal released is noted, which suggests a good stability in a physiological environment. On the other hand, an assessment was not carried out at body temperature, in fact the release was carried out at 25 ° C but for higher temperatures they could suggest a greater release with possible undesirable effects on healthy cells. Although the metals used are considered not completely toxic by the FDA for humans, higher doses of metal for more or less prolonged times in the body could cause side effects.

Cu-MOF was highlighted: it possesses considerable cytotoxic capabilities against cancer cells, on the other hand there is always the problem of being able to bring this compound to a broader level.

The synthesized Cu-MOF has certainly shown an improved power of the effect on cancer cells, but the problem still remains as to how one could act to select the cancer cell from the non-cancer one? Evaluating the effect on tumor cells taken individually is not enough to understand the interaction in a more complex environment.

In conclusion, the work performed potentially presents useful data for understanding the activity of the MOF in a given physiological environment, but some parameters such as temperature need to be better described. Despite the improved IC50, there is a lack of data necessary to guarantee the specificity of the MOF on the tumor cell so that this Cu-MOF can differentiate itself from other drugs used in chemotherapy. So I suggest major revisions before any further consideration for publication.

Author Response

Manuscript ID: pharmaceutics-1552540
Type of manuscript: Article
Title: Therapeutic Application of Metal–Organic Frameworks Composed of 
Copper, Cobalt, and Zinc: Their Anticancer Activity and Mechanism
Authors: Ihn Han, Seung Ah Choi, Do Nam Lee *

Review 2.

Comments and suggestions for authors

The article “Therapeutic Application of Metal-Organic Frameworks Composed of Copper, Cobalt, and Zinc. Their Anticancer Activity and Mechanism” by I. Han, A. Choi, and D. N. Lee deals with the treatment of tumor cells using MOF and is based on well-known state of the art.

The idea of MOF in drug delivery is very compelling, but many aspects still need to be improved before it can really be applied. In particular, the work describes three MOFs based on copper, cobalt, and zinc. Each metal is linked by glutamate forming a 2D sheet structure and then these sheets formed a bridge with the bpa, resulting in the 3D structure.

From the values obtained on the stability studies, Cu-MOF was stable for 24h mainly in deionized water, Co-MOF both in water and in DPBS and Zn-MOF mainly in DPBS. Evaluating in particular the results obtained on the release of metal in aqueous environment and in DPBS, a very low quantity of metal release is noted, which suggests a good stability in a physiological environment. On the other hand, as assessment was not carried out at body temperatures. In fact, the released was carried out at 25°C but for higher temperatures they could suggest a greater release with possible undesirable effects on healthy cells. Although the metals used are considered not completely toxic by the FDA for humans, higher doses of metal for more or less prolonged times in the body could cause side effects. Cu-MOF was highlighted: it possesses considerable cytotoxic capabilities against cancer cells, on the other hand there is always the problem of being able to bring this compound to a boader level.

The synthesized Cu-MOF has certainly shown an improved power of the effect on cancer cells, but the problem still remains as to how one could act to select the cancer cell from the non-cancer one? Evaluating the effects on tumor cells taken individually is not enough to understand the interaction in a more complex environment.

In conclusion, the work performed potentially presents useful data for understanding the activity of the MOF in a given physiological environment, but some parameters such as the temperature need to be better described. Despite the improved IC 50, there is a lack of data necessary to guarantee the specificity of the MOF on the tumor cell so that this Cu-MOF can differentiate itself from other drugs used in chemotherapy. So I suggest major revisions before any further consideration.

Response: We thank the reviewer for valuable comment. Your point is very helpful to us, and we've been thinking about it too.

  • The synthesized Cu-MOF certainly showed the advanced strength of the effect on cancer cells, but as you pointed out, the problem remains as to how it can function in non-cancerous cells to select cancer cells. At first, we previously designed a variety of experiments to solve this. It was performed to evaluate for the occurrence of ROS by metal ions and the redox-reaction with the cell membrane of the bacteria. These previous studies have observed that the cell membrane of microorganism was collapsed on the surface of Cu-MOFs when treated with Cu-MOFs. These results supposed the main cause of the combination of radicals and hydrogen ions in the cell membrane to eliminate hydrogen ions from the cell membrane. As you know that a structured MOF having hydrogen-bonding donor that could be binding with hydrogen on target cell membrane This MOF is an important example of a hydrogen-bonding MOF showing an ability for hydroxyl group recognition. In previous antibacterial application (ref 35), we observed the stability of Cu-MOF from the SEM images of Cu-MOF crystals obtained after bioactive tests, which maintained initial shapes without decomposition during antibacterial test proceeded in PBS at 37°C as following figure.
  • Therefore, Cu-MOF is either confirmed stable at 37°C.
  • In the case of Cu-MOF, the amount of released metal ion was low to affect the bioactivity. We speculate that the proteins and fatty acids in bacterial membranes were susceptible to oxidation or transmembrane potential due to the water-insoluble robust Cu-MOF crystals with active surface metal sites attached to the bacterial surfaces, and this led to inactivation of microbe as previously reported one (W. Zhuang, D. Yuan, J. R. Li, Z. Luo, H.-C. Zhou, S. Bashir and J. Liu, Adv. Healthcare Mater., 2012, 1, 225., R. Hong, T. Y. Kang, C. A. Michels and N. Gadura, Appl. Environ. Microbiol., 2012, 78, 1776)
  •  

-          Figure SEM images of live bacteria and Cu-MOF 2-treated bacteria. The images are obtained under the same magnification (× 20,000).

  • Generally, cancer cells have higher levels of free radicals than normal cells, and this phenomenon is associated with increased metabolism of cancer cells and mitochondrial dysfunction. Increased free radical production in cancer cells brings about biochemical and molecular changes necessary for resistance to chemotherapy for cancer cells along with the development, proliferation, and metastasis of cancer cells. Increasing reactive oxygen species to a level that causes cytotoxicity in cancer cells can be an effective method of removing cancer cells by active oxygen-mediated apoptosis or inhibiting cancer cell resistance [Nature 2015, 520, 57–62. and Free Radic. Biol. Med. 2012, 53, 743-757.]. Therefore, ROS formed during the reduction of metal ions on the surface of metal oxide of MOF are more responsible for toxic effects on the cancer cells than normal cells. Further, ROS are also able to trigger programmed cell death (PCD) of cancer cells as a large amount of active oxygen in a pathological state acts as a toxic substance in the cell. However, a low concentration of active oxygen temporarily generated in a specific area by an external signal is a key factor that regulates cell functions such as cell growth and death as well as immunity. In conclusion, bioactivities and cytotoxicity of MOFs are deduced depending on the unique nature of metal oxide on surface of MOF participation in various reaction at cell wall.
  • A detailed discussion on the specificity of the MOF on the tumor cell was briefly added in the results and discussion section as following

“We have previously investigated the occurrence of ROS by metal ions and the oxidative reaction with the cell membrane of the bacteria [43]. In the previous study, we proposed the main cause of cell membrane collapse might be derived from the combination of radicals and hydrogen ions in the cell membrane to eliminate hydrogen ions from the cell membrane. Combined with cytotoxicity assay and apoptosis assay, where the MOFs were treated and reacted physiological condition at 37°C in CO2 incubator, the reduction cycle between Cu2+ and Cu+ on surface of Cu-MOF attached to cell membrane is accelerated and the pathways that produce reactive oxygen species (ROS) is changed, increasing ROS to a level that causes cytotoxicity to cancer cell and improvement of IC50 [42,49]. Taken together these results, the well-established Cu-MOF activated ROS generation via chemical reaction in aqueous solution, inactivating cancer cells, U87MG, LN229, and SKOV3 as well as E. coli, S. aureus, P. pneumoniae, and P. aeruginosa and MRSA. Therefore, Cu-MOF has a good stability and reusability suitable for biomedical applications.”
